# Tracking moving objects through scattering media via speckle correlations

Y. Jauregui-Sánchez [1] ✉, H. Penketh[1] & J. Bertolotti [1]

Scattering can rapidly degrade our ability to form an optical image, to the point where only speckle-like patterns can be measured. Truly non-invasive imaging through a strongly scattering obstacle is difficult, and usually reliant on a computationally intensive numerical reconstruction. In this work we show that, by combining the cross-correlations of the measured speckle pattern at different times, it is possible to track a moving object with minimal computational effort and over a large field of view.

Most optical imaging systems, including our eyes, use a lens system to reproduce on the detector an appropriately modified (e.g. magnified or filtered) version of the intensity profile on the object plane. This approach requires light to propagate in straight lines, and as soon as the medium between the object and the detector is inhomogeneous, the resulting image is distorted or blurred[1]. If the scattering is weak enough, there is a significant fraction of the light that is left unscattered and that can be used to form a sharp image[2–4], but for strongly scattering media it becomes impossible to filter out the unwanted signal[5]. Knowledge of the exact distortion, the presence of objects of known shape, or strong priors on what the image should look like, allow for the detected image to be corrected and the desired information restored[6–9]. On the other hand, reconstructing an unknown image through a strongly scattering medium is a difficult and still largely open problem[10,11].

A possible approach to this problem is based on the idea that there are universal correlations in the scattered light, that does not depend on the fine details of the medium. In particular, the optical memory effect[12,13] tells us that the speckle pattern we get when coherent light passes through a slab of scattering material[14,15] is strongly correlated with the speckle pattern we get if we illuminate the scattering slab from a different angle[16–19]. This is a powerful tool for imaging, as it gives us information on the light on the hidden side of the scattering medium without ever having to measure anything there. In particular, it allows us to measure the autocorrelation of the hidden object[20–27], which can be inverted using a Gerchberg-Saxton algorithm to obtain the shape of the object[28–30], which is computationally intensive, in the sense that "there are no useful bounds on the number of iterations required to find the solution"[31]. Therefore, if objects in the scene are moving, there is no time to perform the autocorrelation inversion before the scene has changed.

In this work we show that, while the imaging problem is hard, tracking the motion of objects hidden behind a scattering medium is much less so. In particular, the proposed technique has a negligible computational cost, and thus can be performed in real-time, and tracking can be performed well beyond the memory effect range that otherwise limits the field of view[32].

## Results

If we consider an object either emitting or reflecting light hidden behind a strongly scattering layer, each point $\mathbf{x}_o$ will radiate an amount $O(\mathbf{x}_o)$ of light towards the scattering screen (in the following we will assume isotropic emission for simplicity). For a narrow-band enough detection, the light coming from each point will create a speckle pattern $S(\mathbf{x}_o, \mathbf{x}_d)$ at position $\mathbf{x}_d$ on the detector (see Fig. 1). If the spatial coherence is low enough, these speckle patterns will not interfere, but simply sum with each other[33], and the total measured intensity on the detector will be

$$I(\mathbf{x}_d) = \int O(\mathbf{x}_o)S(\mathbf{x}_o,\mathbf{x}_d)d^2\mathbf{x}_o. \tag{1}$$

In this form, there is no obvious way to retrieve $O$, but if we autocorrelate the measured intensity we obtain

$$
\begin{aligned}
[I \star I](\boldsymbol{\Delta x}_d) &= \int I(\mathbf{x}_d)I(\mathbf{x}_d + \boldsymbol{\Delta x}_d)d^2\mathbf{x}_d \\
&= \int \left[ \left( \int O(\mathbf{x}_o)S(\mathbf{x}_o,\mathbf{x}_d)d^2\mathbf{x}_o \right) \left( \int O(\mathbf{y}_o)S(\mathbf{y}_o,\mathbf{x}_d + \boldsymbol{\Delta x}_d)d^2\mathbf{y}_o \right) \right] d^2\mathbf{x}_d \\
&= \int O(\mathbf{x}_o)O(\mathbf{y}_o)([S \star S](\mathbf{x}_o,\mathbf{y}_o,\boldsymbol{\Delta x}_d))d^2\mathbf{x}_o d^2\mathbf{y}_o,
\end{aligned}
\tag{2}
$$

[1]Physics and Astronomy Department, University of Exeter, Stocker Road, Exeter EX4 4QL, UK. ✉e-mail: y.jauregui-sanchez@exeter.ac.uk

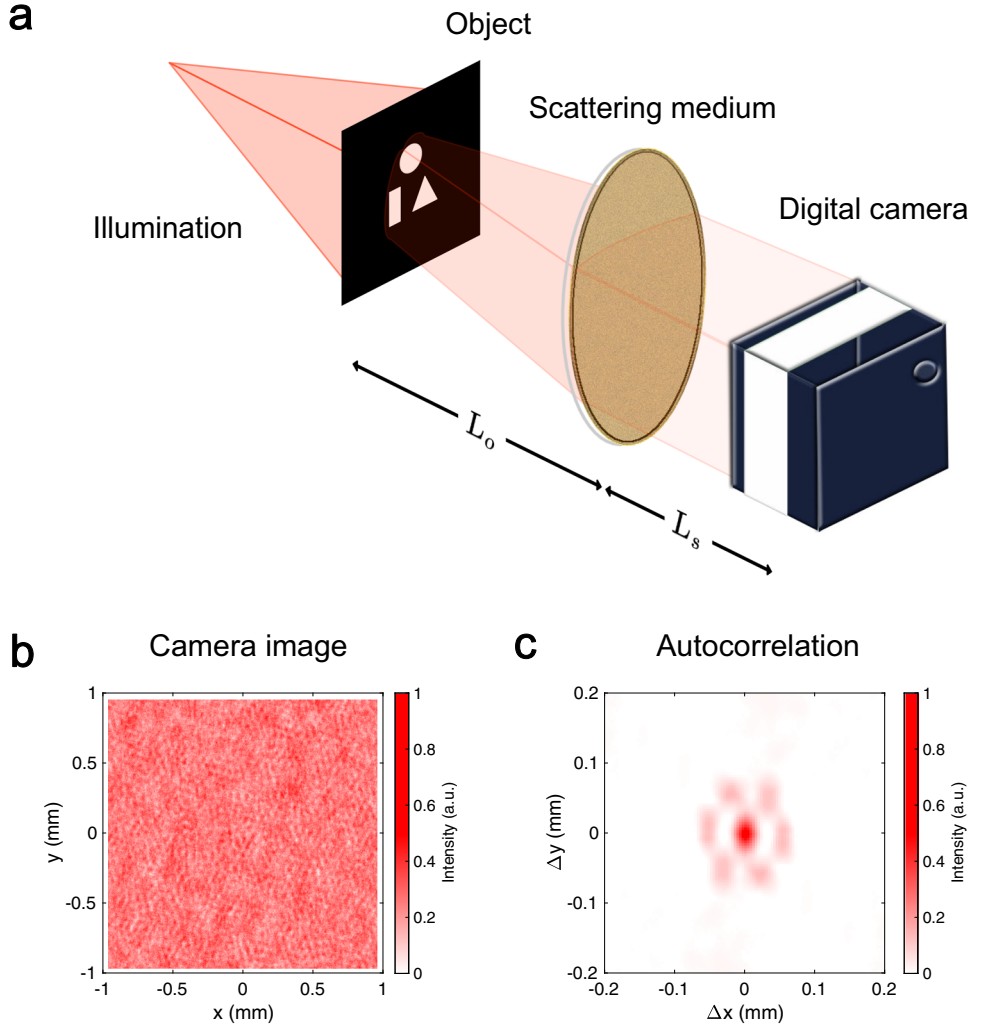

**Fig. 1 | Schematic representation of the experimental setup. a** An object hidden at a distance $L_o$ behind a scattering medium emits light (e.g. because it is illuminated). The scattered light forms a speckle pattern, which is detected by a camera at a distance $L_s$ from the medium (**b**). While the speckle itself looks random, its autocorrelation shows information about the hidden object (**c**). In the present experiment, we used $L_o = 430$ mm and $L_s = 50$ mm.

where $\star$ represents the cross-correlation product. Making an ensemble average $\langle \cdot \rangle$ over disorder, we can rewrite this in a compact form in terms of the speckle correlations $\mathcal{C} = \frac{\langle \delta S \star \delta S \rangle}{\overline{S}^2}$[16]:

$$\langle I \star I \rangle(\mathbf{\Delta x}_\mathrm{d}) \propto [O \star O] \otimes \mathcal{C} \qquad (3)$$

where $\delta S = S - \overline{S}$, $\overline{S}$ is the average intensity of the speckle on the object plane, and $\otimes$ is the convolution product. See the Supplementary Information: S1 for the step by step calculation, and a discussion of the meaning of each term.

Equation (3) holds for any correlation between the speckle generated by incoherent sources at a distance $\mathbf{\Delta x}_\mathrm{o} = \mathbf{x}_\mathrm{o} - \mathbf{y}_\mathrm{o}$, and in particular it works for the optical memory effect[13]:

$$\mathcal{C} \simeq e^{-k^2(\mathbf{\Delta x}_\mathrm{o} - \mathbf{\Delta x}_\mathrm{d})^2 \sigma^2} \cdot \left( \frac{k|\mathbf{\Delta \theta}|L}{\sinh(k|\mathbf{\Delta \theta}|L)} \right)^2, \qquad (4)$$

where we assumed that the illumination is a Gaussian beam with variance $\sigma^2$, $\mathbf{\Delta \theta} \simeq \mathbf{\Delta x}_\mathrm{o}/L_o$ (with $L_o$ the distance between the object plane and the scattering medium, see Fig. 1a), and $L$ the scattering medium thickness. As per Eq. (3), $\mathcal{C}$ behaves like the point spread function of our measurement of $O \star O$. Its first term tells us that the autocorrelation is sharply peaked around $\mathbf{\Delta x}_\mathrm{o} = \mathbf{\Delta x}_\mathrm{d}$, and the second that the correlation

decreases rapidly for large $\mathbf{\Delta x}_\mathrm{o}$, resulting in a finite range where this correlation can be exploited (the isoplanatic patch)[10]. Within this range, we can therefore measure $O \star O$ with a resolution that depends on the wavelength and the width of the illumination beam, but not on the scattering properties of the medium. And while it is not possible to analytically invert an autocorrelation, it is possible to do so numerically under very mild assumptions[24,34,35].

If the scene one wants to image is not static, it is possible to repeat the process above for each desired frame, but since the autocorrelation inversion is a computationally intensive process, prone to get stuck in local minima[30], this can quickly become unwieldy. That said, in many cases one might be satisfied with knowing how the scene moved at any given moment, instead of imaging every single frame[36]. In this case, it is possible to significantly simplify the problem.

If the whole scene moved rigidly, it is possible to reconstruct accurately how much by performing the correlation of the measured light at time $t_1$ with the measured light at time $t_0$. This results in the very same autocorrelation $O \star O$, just shifted by exactly the amount the scene moved[37,38]. But going beyond rigid movements requires some care. Let's assume that the object we want to measure is made of a number of smaller components, each moving independently from the others: $O = \sum_j o_j$. If we try to correlate the intensity at time $t_1$ with the

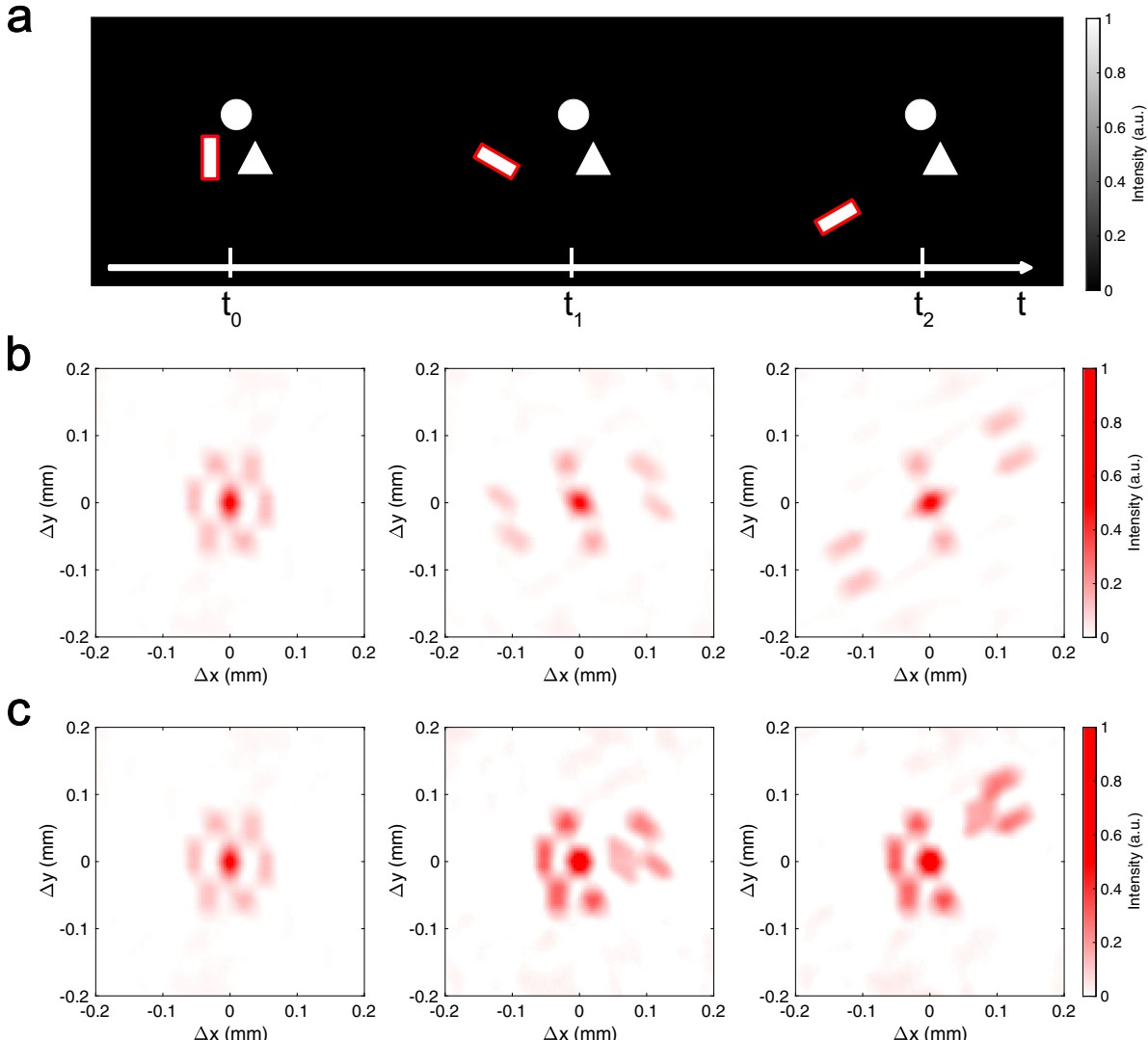

**Fig. 2 | Experimental results of tracking moving objects via speckle correlations in the case of few objects in the scene. a** Displayed intensity patterns on the DMD at different times. The moving object is highlighted in red in the drawing. **b** Autocorrelation of the speckle pattern measured at different times, and **c** cross-correlation between the measured speckle at time $t_0$ and that at time $t_n$ (see Supplementary Video 1).

intensity at time $t_0$ we obtain

$$
\begin{aligned}
I(t_1) \star I(t_0) &\propto \left[ \left( \sum_j o_j(t_1) \right) \star \left( \sum_j o_j(t_0) \right) \right] \otimes \mathcal{C} \\
&= \left[ \sum_j o_j(t_0) \star o_j(t_1) + \sum_{j \neq i} o_j(t_0) \star o_i(t_1) \right] \otimes \mathcal{C}.
\end{aligned}
\tag{5}
$$

The problem with Eq. (5) is that, while the first term tells us how much a given sub-object moved, the second term contains all cross-correlations between all the sub-objects comprising the scene, including information about all the sub-objects that never moved. If we assume for simplicity that only a single sub-object $o_k(t)$ is moving, we can split the above equation into a static and a moving part

$$
I(t_1) \star I(t_0) \propto \left[ \sum_{i,j \neq k} o_i \star o_j + \sum_j o_k(t) \star o_j \right] \otimes \mathcal{C},
\tag{6}
$$

i.e. a part of the cross-correlation will stay static, and another will move following the moving object. If the number of overall sub-

objects $o_j$ is small, the moving part of the autocorrelation is easy to see and interpret (see Fig. 2 and Supplementary Video 1), but if there are many sub-objects the static ones will dominate the cross-correlation, making the moving part difficult to spot. Furthermore, the moving part now contains the cross-correlations with the many non-moving objects, which makes it even harder to parse the image (see Fig. 3b, and Supplementary Video 2). A solution to this problem is to calculate $I(t_1) \star I(t_0) - I(t_1) \star I(t_1)$ instead. If only the $k$th sub-object moves, this reads as

$$
\begin{aligned}
I(t_1) \star I(t_0) - I(t_1) \star I(t_1) &\propto \left[ \sum_j \left[ o_k(t_0) - o_k(t_1) \right] \star o_j \right] \otimes \mathcal{C} \\
&= \left[ \left[ o_k(t_0) - o_k(t_1) \right] \star I(t_1) \right] \otimes \mathcal{C}.
\end{aligned}
\tag{7}
$$

If there are more sub-objects moving, each will yield a term similar to this one. This subtraction has several advantages: first of all, it removes most of the information on the objects that are not moving from the picture, leaving only information on their relative distance from the moving sub-objects. Furthermore, even if two sub-objects are

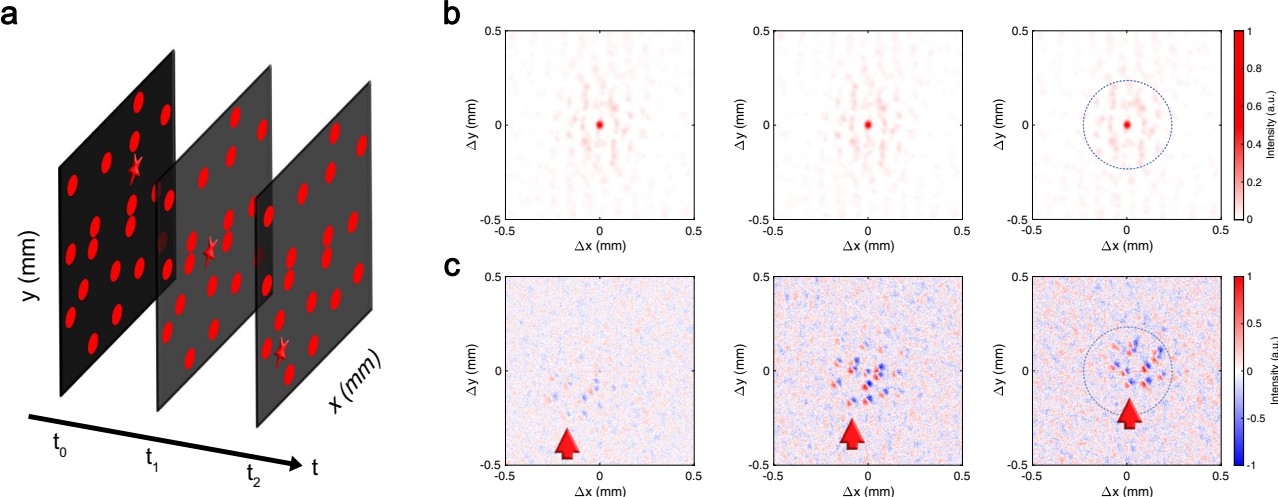

**Fig. 3 | Experimental results of tracking a moving star object over many static dots via speckle correlations. a** Three representative still frames of the moving scene on the DMD, made of a large number of static dots, and a single moving star. **b** The cross-correlation between the measured speckle patterns at $t_0$ and the speckle patterns at successive times. The cross-correlation decreases with distance from the centre due to the finite memory effect range. **c** Plot of $I(t_n) \star I(t_{n-2}) - I(t_n) \star I(t_n)$, where $I(t_n)$ is the measured speckle pattern at frame $n$ (see Supplementary Video 3). The red arrow follows the cross-correlations between one of the static background dots and the moving star. In the first panel, the background dot is at the edge of the memory effect range from the moving star, and it is thus barely visible, becoming brighter as it gets closer to the centre of the picture (i.e. closer to the moving star). The dashed circle in the last panel of **b** and **c** shows the memory range.

too far from each other to be contained within the memory range, we still have information about how the sub-object is moving with respect to the sub-objects near to it. This allows us to track the moving sub-object(s) beyond the memory range (see Fig. 3 and Supplementary Video 2). It is worth noting that one can perform the cross-correlation between the most recent frame and any other previous frames, not necessarily the previous one. This degree of freedom can be used to maximize the visibility of the moving object, depending on the speed of the moving object(s) and the acquisition frame rate. In fact, if the object did not move enough between the frames we correlate, $o_k(t_0) - o_k(t_1)$ in Eq. (7) will be close to zero, producing a weak signal (see Supplementary Video 3).

As an experimental validation of the technique described above, we generated a moving scene on a digital micromirror device (DMD, Texas Instruments DLP9000), and illuminated it with the light from a Red He-Ne laser beam (HNLS008R, Thorlabs), passed through a rotating diffuser (DG20-1500 Ground Glass Diffuser, Thorlabs) to reduce its spatial coherence (see Fig. 1a). Between the DMD and the digital camera (Allied Vision Manta G-125B, Edmund Optics) we place an optical diffuser (DG10-220-MD Ground Glass Diffuser, Thorlabs) to scatter all the light and produce a speckle pattern (see Fig. 1b). The optical diffuser was changed for two layers of adhesive sellotape in the second part of the experiment to limit the memory effect range. It is important to notice that the spatially incoherent illumination is not collimated, so the light reflected by the DMD opens in a wide cone, and no image of the scene is formed on the scattering medium.

If the scene is composed of few objects like in Fig. 2a, the time evolution of the simple autocorrelation of the measured speckle can tell us a lot about how it changed. For instance, in Fig. 2b (and Supplementary Video 1) we can see that something did not move, as the top and bottom parts of the autocorrelation do not change, while something else both got further away and rotated. The problem with looking at the time evolution of the autocorrelation is that, even with such a simple case, it is difficult to extract much more information, unless one has very strong priors on the underlying scene. A significant improvement can be obtained by looking at the cross-correlation of the most recent measured speckle with a previously measured one (in this case, the one measured at $t = t_0$). As shown in Fig. 2c,

the movement of a single object is visible as a part of the autocorrelation detaching. As per Eq. (6), the moving part is composed of the cross-correlation of the moving object with all the other objects (including itself), which in this simple case immediately tells us that the underlying scene is composed of 3 objects, of which only one is moving, and it is both translating and rotating. The amount we see moving in the cross-correlation is linearly proportional to real lateral movement, with a scale factor of $L_s/L_o$. As this method has an infinite depth of field[22], all objects will appear equally in focus irrespectively from their distance. Movement toward or away from the scattering medium can still be inferred from the change in apparent size[25]. Moreover, due to the geometry of the imaging system, the axes of the speckle image, and thus of all autocorrelations and cross-correlations, are reversed relative to the original hidden scene (similarly to a pinhole camera). This is easy to correct by flipping the images if deemed inconvenient. It is also important to notice that, as the autocorrelation of a real function is always centrosymmetric, it is only ever possible to detect relative motion with respect to the position of the objects at the reference time $t_0$, and not an absolute movement.

If the scene becomes more complex, with many objects present (e.g. the object shown if Fig. 3a), the simple cross-correlation becomes difficult to interpret (see Fig. 3b and Supplementary Video 2). In the case of a single (or a few) object(s) moving over an otherwise static background, the static part will dominate the cross-correlations, thus making it impossible to see the movement. If we use Eq. (7) and look at the difference between the cross-correlation and the autocorrelation, we effectively subtract all the static background, leaving just the informative part of the picture, i.e. the cross-correlation of the difference between the moving object at time $t_0$ and $t_1$ with the static background. As $o_k(t_0) - o_k(t_1)$ is negative in the area now occupied by the moving object, and positive in the area it previously occupied, the resulting image (see Fig. 3c) has zero mean, and the negative part shows the direction of movement of the object. It is important to notice that the moving object (or objects, if there is more than one) is always in the centre of the picture, with the (static) background moving around it, essentially showing the movement from the frame of reference of the moving object. Once again, this can be easily corrected numerically if deemed inconvenient. For this experiment,

we changed the optical diffuser for two layers of sellotape to limit the memory effect range, as visible in Fig. 3b, and show that it is possible to track a moving object well beyond the memory range. As shown in Fig. 3c (and Supplementary Video 2), only the cross-correlations with background objects closer to the moving one that the memory effect range are visible (see Eq. (4)). Therefore, as long as the moving object is close enough to at least one static background object, it is possible to track its movements, effectively mosaicing the memory range (see Supplementary Videos 3 and 4).

## Discussion

In this work we show that, while imaging through a strongly scattering medium is still (despite the large amount of work done in the field) a largely unsolved problem, tracking the movement of an object admits a much simpler solution. In particular, an appropriate linear combination of the cross-correlations of the speckle-like images at different times, produces a human-interpretable sequence of frames, which allows the tracking in real time, with minimal computation, and without any need for a reference or a guide star. The motion to be tracked is by no means limited to simple lateral translations, but includes rotations, changes of size (see Supplementary Videos 5 and 7), deformations etc. The main limiting factors of this approach are the memory range and the limited amount of signal available. From Eq. (4) we see that the range of the memory effect decreases exponentially with the thickness $L$. The presence of anisotropic scattering (common in biological media or atmospheric Physics) can increase the viable memory range[19], but for scattering layers much thicker than the wavelength, the requirement that the object must move less than the memory range between two successive frames (otherwise $\mathcal{C}$ will be essentially zero) becomes very restrictive. At the same time if the object did not move enough, the difference in Eq. (7) will be very small. As only a fraction of the light is able to pass through the scattering medium and reach the camera, one risks having to take the difference between two small signals. Therefore, sensitive and fast detectors with a good dynamic range will be required for real-world applications[39].

## Methods

A He-Ne laser beam was expanded and passed through a spinning diffuser to reduce its spatial coherence. The resulting divergent beam was used to illuminate a moving scene generated on a DMD (or a reflecting sample, see Supplementary Information: S3 and Supplementary Video 6). The reflected light passed through a 220-grit ground glass diffuser (DG10-220-MD, Thorlabs), and the resulting speckle-like intensity pattern was collected lenslessly by a CMOS camera with an exposure time of 3 s. To obtain the results in Fig. 3, the scattering layer was changed into two layers of adhesive sellotape, in order to reduce the memory effect range, while keeping a comparable transmission. See Supplementary Information: S2 for further details.

## Data availability

The data used in this study are available at https://doi.org/10.5281/zenodo.6124320.

## Code availability

The MATLAB codes used to process the raw data are available at https://doi.org/10.5281/zenodo.6124320.

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

## Acknowledgements

The research was financially supported by the Engineering and Physical Sciences Research Council (EPSRC) and Dstl via the University Defence Research Collaboration in Signal Processing (EP/S000631/1).

## Author contributions

J.B. developed the original idea and performed the calculations and numerical simulations. Y.J.S. designed and set up the experimental apparatus. Y.J.S. and H.P. performed the experiments and the data analysis. All authors wrote the manuscript.

## Competing interests

The authors declare no competing interests.
