## [Peer Review File · Nature Communications]

Tracking moving objects through scattering media via speckle correlationsREVIEWER COMMENTS

Reviewer #1 (Remarks to the Author):

Imaging through scattering is a fascinating topic as it is a century-old problem with implications in almost any relevant settings, ranging from astronomical to cellular observations. This field has witnessed a remarkable advance after a seminal work (from one of the authors Ref. 21) where one can compute the image of an object buried in a scattering medium based on correlations that persists upon multiple scattering of light, aka, the memory effect. In the original proof-of-concept (Ref. 21), a static object was used, and many others (perhaps all) works that followed up have used static objects. The reason for this bottleneck (in pushing the method to dynamic objects) is simple: recovering the object is computationally intensive, so extending these reconstruction schemes of multiple-frames objects has a high computational cost. Going from a static framework to a dynamic framework would have implications in many fields, in particular biomedical applications as in Life Sciences, almost everything is dynamic often requiring high-speed acquisitions machinery.

To the best of my knowledge, Jauregui-Sanchez et al presents the first idea how to tackle this challenge. They have realised that, while the computational cost of retrieving an image is still unchallenged, the relative movement among different parts of an object can be faithfully and simply retrieved. They were able to demonstrate that a differential approach in the correlations of the speckles reveals subtle relative position movements. Intuitively, differential measurements are not novel but finding a rigorous model that retrieves absolute physical properties is far from trivial. They put up an analytical model to show the effect rigorously and also demonstrate it using a simple optical setup. Again, while this is not what we intuitively expect (the norm is to see the dynamic object as it changes), this work will certainly be a cornerstone step to spark a series of novel ideas on this specific challenge. A very nice extra bonus of their method is the ability to retrieve the relative distance of the moving object even if it moves beyond the memory effect range. For these reasons, I believe the work fits nicely the Nature Communication audience, in particular the readers of imaging in complex media. Nevertheless, I put up below some minor improvements/questions the authors should consider for a revised manuscript

1) The premises for the work is that current paradigms in retrieving the object are computationally intensive. It would be good if the authors could provide a bit more in-depth discussion (or brief explanation) or at least point to a reference that analysed the computational complexity of the problem. There is a vague reference to a 1982 paper (ref. 28), but this is clearly outdated as the phase-retrieval field has considerably advanced in the last decades.

2) One thing that is not clear to me is that there is a need of “ensemble average over disorder”. The authors should discuss the importance of this averaging process over the time scale for the frame change.

3) Equations 5-7 have a function C convoluted with other terms: these equations implies single realisation, whereas C arises from ensemble averaging (judging from the SI derivation). Can the authors clarify the meaning of this?

4) The resilience to measure displacement beyond the memory effect range is really a nice feat. I'd suggest the authors to graphically depict or mention the range of the correlation when presenting the results of figure 3.

5) L156-159: it is not clear where the authors mean if there is a movement in the direction of the scattering medium.

Reviewer #2 (Remarks to the Author):

The authors demonstrate a cross-correlation method to track a moving object with minimal

computation over a large field of view. Although the idea of tracking moving targets behind scattering layers via speckle correlation is not totally new, the authors develop this method to a new level with new application scenario. The theoretical deduction and experimental verification are convincing. I believe that, on the whole, this work may be interesting and novel enough to deserve publication in Nature Communication. However, I have following suggestions to improve the current manuscript. My comments are listed below.

1. The author discussed the advantages of the measurement technique but their disadvantages are barely mentioned. The illumination scheme, coherent back-light illumination requirement and hence intrusive, the limitation to a thin diffuser so as to generate speckle-like pattern, as well as unsuitability to atmosphere scattering should all be discussed.
2. The speckle pattern with a dynamic object behind a diffuser could also be recovered from point spread function with differential scattering spectra. (Optics Express Vol. 25, Issue 26, 32829-32840 (2017)). The author should mention the related work as it is directly related to the work in this manuscript
3. There are repeated “=” in Equation (2), S2 and S3. Equation (2) can be simplified because the step by step calculation has been shown in the Supplementary Information.
4. A laser beam passing through a spinning diffuser was used as the light source. Why not using a monochromatic LED, what is the main limitation or drawback?
5. Subtraction methods were introduced when the static objects are dominated. The subtraction would be $I(t_1) \star I(t_0) - I(t_1) \star I(t_1)$, $I(t_n) \star I(t_{n-2}) - I(t_n) \star I(t_n)$ or $I(t_n) \star I(t_{n-3}) - I(t_n) \star I(t_n)$. Is there any criterion to summarize or decide which time difference is most appropriate for the subtraction?
6. The author said “The motion to be tracked is by no means limited to simple lateral translations, but includes rotations, changes of size, deformations etc.” However, The manuscript only shows the results of 2D translation. How to apply the present method to rotational or size changing objects?

Reviewer #3 (Remarks to the Author):

This work explores a computational method that is able to track moving targets behind the scattering media. The computational model is revised from a well-known one that takes advantage of “optical memory effect” and uses correlation operation to extract the information. The authors demonstrate their method by using a series of DMD patterns to mimic the moving target.

In general, the manuscript is well-written, and the explanations are clear. It has some novelty, but still need more convincing and practical experimental results to support the usefulness of such method. I have some major concerns below.

1. Besides using DMD to “mimic” only one moving target, the authors need to show a more realistic scenario of using such method to implement tracking. Such scenarios can be some live biological samples with fluorescence emission, particles in fluid, or other more realistic ones that the authors can think of. At least, similar scenarios shown in supplement Fig. S2 with more complicated structures will be needed.

For DMD mimic demonstration, showing multiple moving targets and evaluating its performance will be helpful.

In supplement Fig. S2, why the authors don't perform the operation of Eq.7 in the main text, rather only perform the operation of Eq. 6 in the main text? If the performance of this case is good, can the authors move the results to the main text?

2. When there are too many static objects emitting light, multiple speckle patterns add up on the sensor and the measurement dynamic range will become an issue, as the authors already mention it in the supplement S1 round line 29.

Could the authors provide more analysis on the camera dynamic range requirement? If using a standard camera with 8-bit bit depth, what will be the requirement for the samples? How noise (shot noise, camera noise, etc) affects the performance? Since this is a statistical method, how many pixels

will be needed to have the reconstruction results lie within the 90% confidence interval?

3. The authors should cite this reference: Milad I. Akhlaghi and Aristide Dogariu, "Tracking hidden objects using stochastic probing," *Optica* 4, 447-453 (2017)

In this reference, the authors from CREOL can track a target in 3D within a fully enclosed scattering box using a statistic method. How this manuscript outperforms the above reference?

4. In Fig. 3c, from left to right, the "correlation speckle group" moves from lower left to top right. However, since the authors are doing relative operation, I expect the "correlation speckle group" to stay at the same location as the operation is based on all the frames that have the same offset of 2. In other words, the operation doesn't know the history of the images, the only temporal information it has is the frame offset between the two frames that undergo the correlation operation. Then, why Fig. 3c shows the entire trajectory (or entire history) of the "star" object? Does the authors do post processing?

Reply Letter: Tracking moving objects through scattering media via speckle correlations

Y. Jauregui-Sánchez, H. Penketh, and J. Bertolotti

Physics and Astronomy Department, University of Exeter,
Stocker Road, Exeter EX4 4QL, UK

Corresponding author: y.jauregui-sanchez@exeter.ac.uk

General answer to all reviewers: The authors would like to thank the reviewers for the helpful suggestions, many of which helped us to improve the manuscript. The comments of each of the reviewers and the response to each of them are detailed below.

Reviewer 1: Imaging through scattering is a fascinating topic as it is a century-old problem with implications in almost any relevant setting, ranging from astronomical to cellular observations. This field has witnessed a remarkable advance after a seminal work (by one of the authors Ref. 21) where one can compute the image of an object buried in a scattering medium based on correlations that persist upon multiple scattering of light, aka, the memory effect. In the original proof-of-concept (Ref. 21), a static object was used, and many others (perhaps all) works that followed up have used static ob-

jects. The reason for this bottleneck (in pushing the method to dynamic objects) is simple: recovering the object is computationally intensive, so extending these reconstruction schemes of multiple-frame objects has a high computational cost. Going from a static framework to a dynamic framework would have implications in many fields, in particular, biomedical applications as in Life Sciences, almost everything is dynamic often requiring high-speed acquisitions machinery.

To the best of my knowledge, Jauregui-Sanchez et al. present the first idea on how to tackle this challenge. They have realised that, while the computational cost of retrieving an image is still unchallenged, the relative movement among different parts of an object can be faithfully and simply retrieved. They were able to demonstrate that a differential approach in the correlations of the speckles reveals subtle relative position movements. Intuitively, differential measurements are not novel but finding a rigorous model that retrieves absolute physical properties is far from trivial. They put up an analytical model to show the effect rigorously and also demonstrate it using a simple optical setup. Again, while this is not what we intuitively expect (the norm is to see the dynamic object as it changes), this work will certainly be a cornerstone step to spark a series of novel ideas on this specific challenge. A very nice extra bonus of their method is the ability to retrieve the relative distance of the moving object even if it moves beyond the memory effect range. For these reasons, I believe the work fits nicely the Nature Communication audience, in particular the readers of imaging in complex media. Nevertheless, I put up below some minor improvements/questions the authors should consider for a revised manuscript. **Authors:** We thank the reviewer for the excellent summary of our work and its place in the broader context of imaging research.

Q1. The premise for the work is that current paradigms in retrieving the object are computationally intensive. It would be good if the authors could provide a bit more in-depth discussion (or brief explanation) or at least point to a reference that analysed the computational complexity of the problem. There is a vague reference to a 1982 paper (Ref. 28), but this is clearly outdated as the phase-retrieval field has considerably advanced in the last decades.

Authors: We agree with the reviewer that we should have done a better job at presenting the current state of the art in phase retrieval. Following the reviewer’s suggestion we added the following paragraph in the main article [L39-42]: “[20-27], which can be inverted using a Gerchberg-Saxton algorithm to obtain the shape of the object [28-30], which is computationally intensive, in the sense that “there are no useful bounds on the number of iterations required to find the solution” [31]. Therefore,”.

Reference added: [31] V. Elser, “Phase retrieval by iterated projections,” *J. Opt. Soc. Am. A.* **20**, 40–55, 2003.

Q2. One thing that is not clear to me is that there is a need for “ensemble average over disorder”. The authors should discuss the importance of this averaging process over the time scale for the frame change.

Authors: When calculating speckle correlations (in this case, the optical memory effect) it is common to calculate the ensemble averaged correlation to avoid having to deal with a particular shape of the speckle (see e.g. E. Akkermans and G. Montambaux, Cambridge University Press, 2007). In the experiment we do not perform any ensemble average (which only appears in equation 3), and the result is a grainy background, clearly visible in Fig. 3,

but also in Fig. 2, which is not described by the equations. To clarify the point we added a sub-section in the Supplementary Information (**S1.1**).

Q3. Equations 5-7 have a function C convoluted with other terms: these equations imply single realisation, whereas C arises from ensemble averaging (judging from the SI derivation). Can the authors clarify the meaning of this?

Authors: As mentioned in Q2, the theoretical analysis is done for the averaged correlation, but the experiments are not. The result is that all experiments show a grainy background that is not described by the theory. To clarify the point we added a sub-section in the Supplementary Information (**S1.1**).

Q4. The resilience to measure displacement beyond the memory effect range is really a nice feat. I'd suggest the authors graphically depict or mention the range of the correlation when presenting the results in figure 3.

Authors: We agree with the reviewer that adding an indication of how wide the memory range was in our measurements is helpful for the reader, so we modified Fig. 3 and **Supplementary Video 2**, adding a dashed circle at the approximate limit of the memory range.

Q5. L156-159: it is not clear where the authors mean if there is a movement in the direction of the scattering medium.

Authors: Imaging using the optical memory effect has an infinite depth of field, so all objects will be in focus irrespectively of how close or far away they are (although further away objects will likely look dimmer). As a result this approach has no "sectioning" capability, but it is possible to detect

movement in the direction of the scattering medium by the change in the apparent size of the object (closer objects will look larger). To make this point clearer, we have changed the sentences in the main article as follows [L159-162]: “As this method has an infinite depth of field [22], all objects will appear equally in focus irrespectively from their distance. Movement toward or away from the scattering medium can still be inferred from the change in apparent size [25]”. We also added a section in the Supplementary Information where we discuss this point (**S3.1**).

Reviewer 2: The authors demonstrate a cross-correlation method to track a moving object with minimal computation over a large field of view. Although the idea of tracking moving targets behind scattering layers via speckle correlation is not totally new, the authors develop this method to a new level with a new application scenario. The theoretical deduction and experimental verification are convincing. I believe that, on the whole, this work may be interesting and novel enough to deserve publication in Nature Communication. However, I have the following suggestions to improve the current manuscript. My comments are listed below.

Q1. The author discussed the advantages of the measurement technique but their disadvantages are barely mentioned. The illumination scheme, coherent back-light illumination requirement and hence intrusive, the limitation to a thin diffuser so as to generate a speckle-like pattern, as well as unsuitability to atmosphere scattering should all be discussed.

Authors: We agree with the reviewer that we should have been more clear about the disadvantages and limitations for all the readers who are not specialists in the field. To remedy we have been added the following

paragraph to the main text [L205-217]: “The main limiting factors of this approach are the memory range and the limited amount of signal available. From equation 4 we see that the range of the memory effect decreases exponentially with the thickness L . The presence of anisotropic scattering (common in biological media or atmospheric Physics) can increase the viable memory range [19], but for scattering layers much thicker than the wavelength, the requirement that the object must move less than the memory range between two successive frames (otherwise \mathcal{C} will be essentially zero) becomes very restrictive. At the same time if the object did not move enough, the difference in equation 7 will be very small. As only a fraction of the light is able to pass through the scattering medium and reach the camera, one risks having to take the difference between two small signals. Therefore, sensitive and fast detectors with a good dynamic range will be required for real-world applications [39]”

Q2. The speckle pattern with a dynamic object behind a diffuser could also be recovered from the point spread function with differential scattering spectra. (Optics Express Vol. 25, Issue 26, 32829-32840, 2017). The author should mention the related work as it is directly related to the work in this manuscript.

Authors: We are aware of the paper the reviewer refers to, but since it uses a known object to retrieve the point spread function, not dissimilarly to how it is done with a guide star, while our method is referenceless and does not attempt to recover the point spread function at all, we did not think to include it among our references. Nevertheless, we agree with the reviewer that it is a good article, and we have now included it (it is now reference [9] in the manuscript).

Q3. There are repeated “=” in the equation (2), S2 and S3. Equation (2) can be simplified because the step by step calculation has been shown in the Supplementary Information.

Authors: We understand where the reviewer is coming from with this comment, but this was a precise choice when writing the manuscript. Most people will not read the supplementary information, so we feel that having at least the main steps clearly spelled out in the main text is a feature, not a bug.

Q4. A laser beam passing through a spinning diffuser was used as the light source. Why not use a monochromatic LED, what is the main limitation or drawback?

Authors: The only reason behind the use of a laser as a light source was simplicity when designing this proof-of-concept experiment. The use of a laser with a separate spinning diffuser offers greater flexibility in the illumination (angular distribution, spot size) of the object when compared to the fixed emitting area of an LED chip. In a watt-for-watt comparison with an LED which may require further filtering to match the bandwidth of the diffuser, the use of a laser provides a higher signal-to-noise ratio, thus making the detection easier. Furthermore, we already know that this kind of correlation-based methods work when using incoherent light (see ref. 23). We are currently working on a follow-up study, where the illumination is provided by a LED, and the objects moving are physical (macroscopic) objects instead of being displayed on the DMD. That said, we believe that the simple experiment we describe is sufficient to support all of our claims.

Q5. Subtraction methods were introduced when the static objects are dominated. The subtraction would be $I(t_1) \star I(t_0) - I(t_1) \star I(t_1)$, $I(t_n) \star I(t_n - 2) - I(t_n) \star I(t_n)$ or $I(t_n) \star I(t_n - 3) - I(t_n) \star I(t_n)$. Is there any criterion to summarize or decide which time difference is most appropriate for the subtraction?

Authors: If the distance between the frames used is too small, the object will not have time to move much, so the two terms will be almost identical, resulting in a difference smaller than the detection noise. If the distance between the frames used is too large, it will get harder and harder to properly localize the object. And if the object moved more than the range of the memory effect between the frames, no tracking is possible at all. In the main text, we mention [L123-127]: “This degree of freedom can be used to maximize the visibility of the moving object, depending on the speed of the moving object(s) and the acquisition frame rate. In fact, if the object did not move enough between the frames we correlate, $o_k(t_0) - o_k(t_1)$ in equation 7 will be close to zero, producing a weak signal”. Moreover, in the newly added **Supplementary Video 7** we show a simulation where, among the other things, the object moves at variable speed. There it is clearly visible that if the object did not move enough between frames almost no signal can be seen.

Q6. The author said, “The motion to be tracked is by no means limited to simple lateral translations, but includes rotations, changes of size, deformations etc.” However, The manuscript only shows the results of 2D translation. How to apply the present method to rotational or size changing objects?

Authors: Fig. 2 shows an object rotating, but we agree that it would be better to have somewhere an example of an application of equation 7 for an

object moving beyond simple translations. To do that we added 2 new supplementary videos (**Supplementary Video 5** and **7**), showing the effect of an object changing size in both an experiment and a simulation. Moreover, to clarify this point, we added a sub-section in the Supplementary Information (**S3.1**).

Reviewer 3: This work explores a computational method that is able to track moving targets behind the scattering media. The computational model is revised from a well-known one that takes advantage of the “optical memory effect” and uses correlation operation to extract the information. The authors demonstrate their method by using a series of DMD patterns to mimic the moving target. In general, the manuscript is well-written, and the explanations are clear. It has some novelty but still needs more convincing and practical experimental results to support the use of such a method. I have some major concerns below.

Q1. Besides using DMD to “mimic” only one moving target, the authors need to show a more realistic scenario of using such method to implement tracking. Such scenarios can be some live biological samples with fluorescence emission, particles in a fluid, or other more realistic ones that the authors can think of. At least, similar scenarios shown in supplement Fig. S2 with more complicated structures will be needed. **Authors:** While we agree with the reviewer that showing this method in a real-life, bio-imaging scenario would be great, this would take many months (possibly years) of work, and therefore we find this request unreasonable. The experiment presented here is a proof of principle (which helps to keep the explanation simple and clear), but we think it supports all of our claims. Future work will move in the di-

rection suggested by the reviewer, but there are a number of issues to work out before we get there.

For DMD mimic demonstration, showing multiple moving targets and evaluating its performance will be helpful. **Authors:** Disentangling the effect of multiple objects' motion remains an open problem, and will be the subject of future study.

In supplement Fig. S2, why the authors don't perform the operation of Eq. 7 in the main text, but rather only perform the operation of Eq. 6 in the main text? If the performance of this case is good, can the authors move the results to the main text? **Authors:** Following the reviewer's suggestion, we modified Fig. S2 and added **Supplementary Video 6** to show the results of applying equation 7.

Q2. When there are too many static objects emitting light, multiple speckle patterns add up on the sensor and the measurement dynamic range will become an issue, as the authors already mention in the supplement S1 round line 29. Could the authors provide more analysis on the camera dynamic range requirement? If using a standard camera with 8-bit bit depth, what will be the requirement for the samples? How noise (shot noise, camera noise, etc) affects the performance? Since this is a statistical method, how many pixels will be needed to have the reconstruction results lie within the 90% confidence interval?

Authors: We thank the reviewer for pointing out that we were not very clear on this point. The $\overline{S}^2 A \|O\|^2$ term indeed produces a large background, but it doesn't do that in the image one measures, but in its autocorrelation. Therefore the dynamic range of the camera is not crucial, as long as the measured speckle pattern is well resolved. In all our measurements we used

a camera with 8-bit dynamic range, and while in certain measurements its sensitivity was not as good as one might have wanted, the dynamic range was never a problem. To make things more clear we removed the confusing sentence the reviewer mentions, and added a section in the supplementary information (section **S2.1**) where we discuss the effects of noise and sensitivity.

Q3. The authors should cite this reference: Milad I. Akhlaghi and Aristide Dogariu, "Tracking hidden objects using stochastic probing," *Optica* 4, 447-453 (2017). In this reference, the authors from CREOL can track a target in 3D within a fully enclosed scattering box using a statistical method. How this manuscript outperforms the above reference?

Authors: The two methods work in very different conditions, and it is impossible to directly compare them and decide which outperforms the other. In particular, the method described in the Akhlaghi paper measures the instantaneous velocity of the moving object, similarly to what is done in diffuse wave spectroscopy, while we measure the absolute distance between the moving object and all the other objects. In particular, while in the Akhlaghi paper the presence of non-moving objects would be disruptive, our method needs non-moving objects to be present to work at all. That said, we agree that the one suggested by the reviewer is a very good and important paper, so we have now included it as reference 36.

Q4. In Fig. 3c, from left to right, the "correlation speckle group" moves from lower left to top right. However, since the authors are doing the relative operation, I expect the "correlation speckle group" to stay at the same location as the operation is based on all the frames that have the same offset of 2. In other words, the operation doesn't know the history of the images,

the only temporal information it has is the frame offset between the two frames that undergo the correlation operation. Then, why Fig. 3c shows the entire trajectory (or entire history) of the “star” object? Do the authors do post-processing?

Authors: Our method can only measure relative distances between the objects, and equation 7 is designed to keep the information about the distances between the moving object(s) and the stationary ones (everything else is cancelled by the subtraction). While the moving object is moving, the relative distances with all the stationary objects change, and each cross-correlation operation (i.e. each application of equation 7) capture a snapshot of how the relative distances changed between the two chosen frames. Each red-blue couple shows how the distance between the moving object and one of the non-moving object changed in the time between the two chosen frames, thus showing at the same time the average distance (the moving object is fixed in the centre by construction), and the velocity of the movement. We want to stress that in Fig. 3c no trajectory is showed, although as clearly visible in the supplementary videos, our brain is very good at automatically interpolate to estimate a trajectory. Finally, we would like to mention that no post-processing was performed on Fig. 3 beyond that described in the second paragraph of Section S2 Experimental setup (and the addition of the red arrows).

REVIEWERS' COMMENTS

Reviewer #1 (Remarks to the Author):

The authors have addressed all points raised by the reviewers convincingly. The updated manuscript includes the most pertinent points that did not undermine their main message: imaging dynamic scattering media with realistic computational recovery strategies. I believe this paper, despite being a proof-of-concept, has all merits to be a cornerstone paper for dynamic imaging through scattering media, and therefore I suggest publication in Nature Communications. On a side note, I hope the authors do push the method to more realistic biological specimens found in microscopy, but that certainly will require a lot more efforts to get it done.

Reviewer #2 (Remarks to the Author):

I regard that authors sufficiently addressed my concern. I recommend the publication of the paper as it is.

Reviewer #3 (Remarks to the Author):

The authors addressed my technical questions in the review comments. I do agree that this work has its novelty that can realize the object tracking beyond the “memory effect” range, and the proof-of-concept experiments do support the claims in the manuscript. Nevertheless, I also hope the authors could implement this method in a more practical manner in the future, if not in the current manuscript.